# Biomimetic high performance artificial muscle built on sacrificial coordination network and mechanical training process

Zhikai Tu[1,4], Weifeng Liu [1,4✉], Jin Wang[2], Xueqing Qiu[3✉], Jinhao Huang[1], Jinxing Li[1] & Hongming Lou[1]

Artificial muscle materials promise incredible applications in actuators, robotics and medical apparatus, yet the ability to mimic the full characteristics of skeletal muscles into synthetic materials remains a huge challenge. Herein, inspired by the dynamic sacrificial bonds in biomaterials and the self-strengthening of skeletal muscles by physical exercise, high performance artificial muscle material is prepared by rearrangement of sacrificial coordination bonds in the polyolefin elastomer via a repetitive mechanical training process. Biomass lignin is incorporated as a green reinforcer for the construction of interfacial coordination bonds. The prepared artificial muscle material exhibits high actuation strain (>40%), high actuation stress (1.5 MPa) which can lift more than 10,000 times its own weight with 30% strain, characteristics of excellent self-strengthening by mechanical training, strain-adaptive stiffening, and heat/electric programmable actuation performance. In this work, we show a facile strategy for the fabrication of intelligent materials using easily available raw materials.

[1] School of Chemistry and Chemical Engineering, Guangdong Engineering Research Center for Green Fine Chemicals, South China University of Technology, Guangzhou, P. R. China. [2] The National Engineering Research Center of Novel Equipment for Polymer Processing, School of Mechanical & Automotive Engineering, South China University of Technology, Guangzhou, P. R. China. [3] School of Chemical Engineering and Light Industry, Guangdong University of Technology, Guangzhou, P. R. China. [4] These authors contributed equally: Zhikai Tu, Weifeng Liu. ✉email: weifengliu@scut.edu.cn; qxq@gdut.edu.cn

The majority of biological load-bearing tissues including skeletal muscles present typical J-shaped stress–strain curves with their initially soft elastic modulus rapidly increased by orders of magnitude during deformation (Supplementary Fig. 1), which is also known as the strain-adaptive stiffening performance[1–3]. Specifically, skeletal muscles can self-strengthen after physical exercise, because of the destruction and reconstruction process of the muscle fibrils upon mechanical training[4]. Moreover, they can intelligently convert adenosine triphosphate (ATP) as chemical energy source into macroscopic motions to adapt the surrounding environment according to the electrical signals sent by the nervous system[5]. These combinational characteristics of skeletal muscles are indispensable for the survival of living organisms in complex environment, which has triggered tremendous interest for the development of artificial muscle materials.

The bio-inspired artificial muscle materials are supposed to be able to convert external stimulus energy into macroscopic motions for a wide scope of potential applications such as actuators, robotics, and medical apparatus[6]. Under the stimulation of external heat, light, electricity etc., the difference in thermal expansion ratios[7], the liquid crystalline-isotropic phase transition[8] or the change of electric field[9] inside the artificial muscle materials would result in macroscopic reversible deformation. For example, a tendril-like fiber-based artificial muscle material was recently fabricated by thermal drawing from the hybrid of cyclic olefin copolymer elastomer and high-density polyethylene[7]. The macroscopic reversible deformation originating from the difference of two polymers in thermal expansion coefficients endowed the material with rapid thermal actuation[7]. Although these previously reported artificial muscle materials could convert external stimulus energy into macroscopic motions, the basic mechanical performance did not fit the features of skeletal muscles such as self-strengthening by mechanical training and strain-adaptive stiffening. Some recent studies have reported self-strengthening hydrogels by repetitive mechanical training strategy[10–12]. However, these materials with self-strengthening characteristics lack the intelligent features such as precise modulation and programmable control of actuation in response to external stimulus. Until now, integrating the full characteristics of skeletal muscles such as self-strengthening by mechanical training, strain-adaptive stiffening and intelligent actuation upon external stimulus, into a single synthetic material remains to be a huge challenge.

Recently, the excellent mechanical performance of some biological materials such as mussel byssus has been partly attributed to the reinforcing mechanism of dynamic sacrificial bonds including the sacrificial coordination bonds and hydrogen bonds[13,14]. Many efforts have been devoted to incorporate dynamic sacrificial bonds into polymeric materials to enhance the mechanical properties[15–21]. For instance, Filippidi et al. found that via introducing $Fe^{3+}$-catechol coordination bonds into the lightly cross-linked elastic epoxy network, the tensile strength and toughness of the elastomer was increased by 58 times and 92 times, respectively[22]. Nevertheless, the coordination bonds in these systems were randomly distributed in the polymer matrix, followed by irregular rupture during the deformation course, giving the material limited strain-adaptive stiffening performance.

In this work, inspired by the dynamic sacrificial bonds in biomaterials and the self-strengthening mechanism of skeletal muscles by physical exercise, we propose a strategy to combine the superiority of sacrificial coordination bonds with mechanical training process for construction of high performance artificial muscle materials. The sacrificial coordination bonds are first introduced into a poly(ethylene-propylene-diene monomer) (EPDM) elastomer via incorporating biomass lignin as natural green reinforcer, followed by a mechanical training process.

Similar to the destruction and reconstruction process of the muscle fibrils upon physical exercise, through repetitive pre-stretching of mechanical training process, the rupture and reconstruction of randomly distributed coordination bonds prompt the orientation of chain segments along the pre-stretching direction. After mechanical training, the newly regenerated coordination bonds along the pre-stretching direction can not only prompt strain-induced crystallization (SIC) more efficiently, but also break at a relatively concentrated strain range, which imparts a rapidly increasing modulus. The prepared EPDM-based artificial muscle material exhibits excellent self-strengthening effect by mechanical training and strong strain-adaptive stiffening performance. Meanwhile, it can actuate programmable reciprocating motions under the stimulation of heat and electricity. The EPDM-based artificial muscle material in this work successfully mimics the comprehensive mechanical characteristics of skeletal muscles via a facile approach built on sacrificial coordination bonds and mechanical training process, laying a foundation for large-scale fabrication of artificial muscle materials.

## Results

**Preparation and mechanical performance of the EPDM-based artificial muscle material.** A three-step compounding process in one internal mixer was conducted to incorporate Zn-based coordination bonds into the EPDM/lignin composite system, as shown in Fig. 1a. EPDM was first blended with lignin, then mixed with zinc dimethacrylate (ZDMA), and finally with vulcanizing agents. The sample was named as LxZy@m%, with "L" standing for lignin, "Z" for ZDMA, and the numbers "x" and "y" for the parts per hundred of EPDM (100 phr), "@" for mechanical training, and the number "m" for the mechanical training strain. The biomass lignin is the largest aromatic biopolymer in nature and rich in oxygen-containing polar groups (Supplementary Fig. 2), which can efficiently function as natural ligands for the construction of interfacial coordination bonds. During the vulcanization process, the unsaturated double bonds in EPDM reacted with the methacrylate groups in ZDMA, generating the ZDMA-modified EPDM with functional carboxylate groups (Fig. 1b and Supplementary Fig. 3)[23]. The Zn-based coordination bonds could form between the carboxylate groups grafted on EPDM backbones and the polar functional groups in lignin such as the phenoxy and carboxylate groups[24], which was confirmed by FTIR spectra in Fig. 2a. The absorption peaks at 1276 cm$^{-1}$ were attributed to the C−O stretching vibration peaks of aryl groups in lignin, and the signals at 1110 cm$^{-1}$ and 1023 cm$^{-1}$ were assigned to the C–O stretching vibration of alkyl groups in the composites, respectively[25]. All of these peaks had evident increase in intensity, accompanied by the increasing content of ZDMA, indicating the coordination effect between $Zn^{2+}$ and oxygen-containing groups. Additionally, an obvious redshift from 1110 cm$^{-1}$ to 1097 cm$^{-1}$ was also observed as the ZDMA content increased, which further demonstrated the complexation of coordination bonds in composites. The carboxylate groups in ZDMA could also form hydrogen bonds with the polar groups in lignin. Typical engineering stress–strain behaviors of the composites with different content of ZDMA also verified the formation of coordination bonds. As shown in Fig. 2b, the tensile strength and toughness of composites increased gradually with the ZDMA content, due to the dynamic destruction and reconstruction process of coordination bonds, through which they could not only dissipate energy on molecular level, but also eliminate stress concentration, constrain the chain segments, and promote the chain orientation, thus endowing materials with improved strength and toughness[18,20]. Remarkly, L0Z12@0% (14.7 MPa) without lignin exhibited much poorer tensile strength

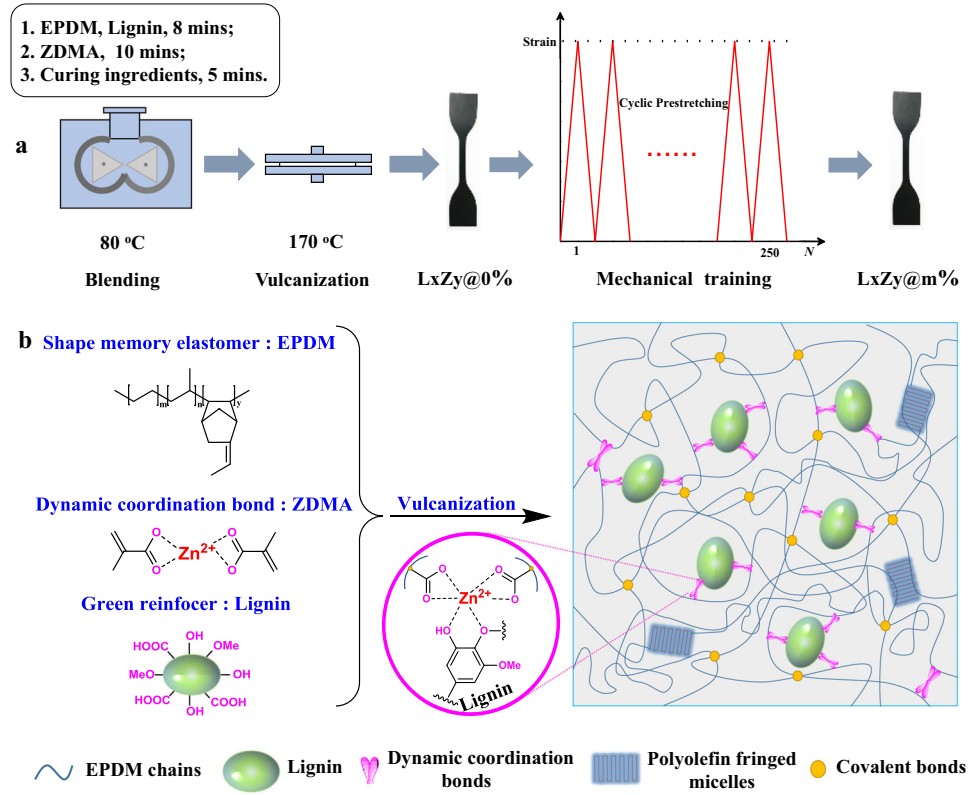

**Fig. 1 The preparation schematic diagram and mechanical perfomance of artificial muscle materials. a** Preparation diagram of EPDM-based artificial muscle materials. **b** Schematic illustration for the interfacial coordination bonds in the EPDM composite.

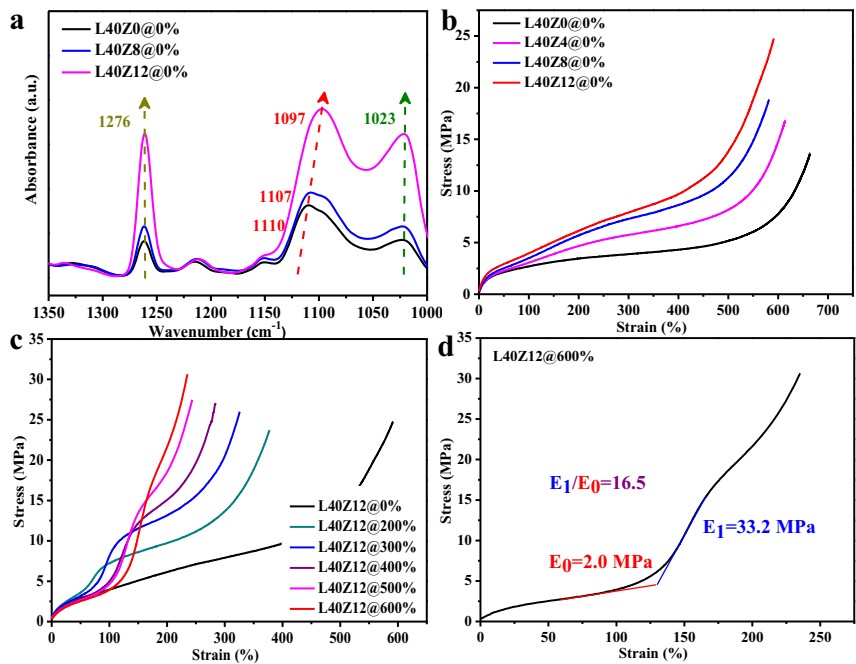

**Fig. 2 Mechanical properties of artifical muscle material. a** The FTIR spectra of EPDM/lignin composites with different content of ZDMA (a.u. means absorbance unit). **b** The engineering stress–strain curves of EPDM/lignin composites with different content of ZDMA. **c** The engineering stress–strain curves of EPDM composites after mechanical training for 250 cycles at the stretching speed of 200 mm/min under different training strain. **d** The tensile stress–strain curve of L40Z12@600% after mechanical training for 250 cycles at 600% strain. Source Data for Fig. 2a–d are available as an associated article file.

compared with L40Z12@0% (24.8 MPa) (Supplementary Fig. 4). The above results proved that reactive coordination bonds and coordinative reinforcer lignin were both necessary to prepare the high-performance EPDM elatomer composite, providing a direction for the utilization of lignin in high-performance elastomer composite, which was a huge challenge in the past.

The EPDM/lignin composite L40Z12@0% was further submitted for mechanical training to mimic the self-strengthening performance of skeletal muscles. The training process was conducted via repeated stretching and unloading (Supplementary Fig. 5), and the corresponding loading–unloading stress–strain curves were illustrated in Supplementary Fig. 6. The engineering stress–strain curves in Fig. 2c demonstrated that the final EPDM/lignin composites exhibited significantly improved strength after mechanical training, just like the self-strengthening effect of skeletal muscles by physical exercise. As the training strain increased from 0 to 600%, the tensile strength at failure of the elastomer composite increased gradually from 24.8 to 30.7 MPa. The stress at 200% strain increased nearly 2.5 times for L40Z12@600% compared with L40Z12@0% (Supplementary Fig. 7a). Increasing the training frequency from 100 to 1000 cycles led to a similar variation tendency in the tensile stress–strain curves (Supplementary Fig. 7b). Under the repetitive pre-stretching of mechanical training, the residual strain of the elastomer also gradually increased with the training strain and training frequency (Supplementary Fig. 7c, d), indicating that a certain part of chain alignment was stabilized by the coordination bonds after mechanical training, which was verified by the relative higher crystallinity of L40Z12@300% than L40Z12@0% shown in Supplementary Fig. 8. The stress relaxation cure also confirmed that only part of rubber chain strands relaxed after mechanical training (Supplementary Fig. 9).

Remarkably, the materials presented typical J-shaped stress–strain curves after mechanical training, conforming to the strain-adaptive stiffening characteristic of soft biological tissues[1–3]. For example, the elastic modulus of the sample L40Z12@600% stiffened 16.5 times from 2.0 MPa at small strain to 33.2 MPa when the strain deformed from 75 to 150%, as shown in Fig. 2d. The self-strengthening effect by mechanical training and the typical strain-adaptive stiffening of the EPDM-based elastomer composites matched well with the mechanical performance of skeletal muscles, demonstrating great potential as artificial muscle materials.

Interestingly, the stress–strain curves of the elastomers after mechanical training revealed a phenomenon of obvious dual-stage enhancement in modulus, which has been rarely reported before. Take the sample L40Z12@300% as an example (Fig. 2c), the first-stage enhancement in modulus occurred in the strain range of 75–150%, followed by the second-stage enhancement. A closer inspection could find that, as the training strain increased from 200 to 600%, the elastic modulus for the both enhancing stages increased gradually, the first-stage enhancement was postponed to a larger strain and the second-stage enhancement was pulled up to a smaller strain, as shown in Fig. 2c, suggesting a special reinforcing mechanism for this EPDM-based artificial muscle material.

**Self-strengthening mechanism by mechanical training**. To investigate the effect of coordination bonds on the mechanical training performance, the control sample L40Z0@300% without ZDMA and the sample L0Z12@300% without lignin were compared by mechanical training at 300% strain for 250 cycles. As shown in Fig. 3a, the tensile strength of L40Z0@300% and L0Z12@300% was much lower than that of the sample L40Z12@300%. Apparently, the dynamic coordination bonds

were constituted of ZDMA providing metal ions and lignin acting as ligands in this EPDM composite system. Without ZDMA, the increase in modulus versus strain (slope of the stress–strain curve) was much weaker in L40Z0@300% compared with L40Z12@300%. While without lignin, the phenomenon of dual-stage enhancement disappeared in the tensile curve of L0Z12@300%. These results demonstrated that the coordination bonds and the green reinforcer lignin played a vital role on the self-strengthening and the strain-adaptive stiffening of the EPDM composite after mechanical training. As the dynamic coordination bonds could increase the interfacial interactions between lignin and polymer matrix[21], the interfacial coordination bonds promoted the internal stress transfer from EPDM matrix to lignin particles, leading to repeated shearing on lignin particles during the mechanical training process, and thus resulting in smaller particle size and better dispersion of lignin in polymer matrix after mechanical training (shown in Supplementary Fig. 10), which also contributed to the self-strengthening of the EPDM composite.

To further reveal the deformation mechanism for the dual-stage enhancement phenomenon, the hysteresis tensile tests at various strains from 50 to 300% were conducted to study the energy dissipation of L40Z12@300% during stretching, as shown in Fig. 3b. It is obvious that the hysteresis loss of L40Z12@300% increased significantly with the applied strain and was larger than that of L40Z12@0% (Supplementary Fig. 11a, b). The hysteresis ratio of L40Z12@300% increased rapidly when the applied strain was between 75 and 150%, but leveled off at strains larger than 150% (Fig. 3b). This suggested that, for L40Z12@300% after mechanical training, the dynamic fracture of coordination bonds might concentrate in the early stage of strain range from 75 to 150%, which led to a rapid growth in the energy dissipation ratio and elastic modulus of the first-stage enhancement. The dynamic fracture of coordination bonds in the strain from 75 to 150% was also demonstrated by the signal variations in the FTIR analysis (shown in Supplementary Fig. 12).

The reason for the second-stage enhancement in the stress–strain curve of L40Z12@300% was attributed to the strain-induced-crystallization (SIC) of the elastomer chain segments after dynamic fracture of coordination bonds. The wide-angle X-ray diffractograms (XRD) of L40Z12@300% at various stretching strains were depicted in Fig. 3c. The sample L40Z12@300% exhibited wide amorphous diffraction peak at 0% strain, with weak reflection of fringed micellar crystals at 2θ of 20.5° hidden in the amorphous diffraction peak. As the strain increased from 50 to 320%, the obvious crystalline peaks at 2θ of 20.5° and 22.8° assigning to the (110) and (200) lattice planes of the polyethylene orthorhombic crystal cell[26], respectively, were observed in L40Z12@300%, and the two peaks both intensified with the strain, which was the typical phenomenon of SIC. As shown in Fig. 3d, the relative crystallinity of L40Z12@300% increased with the strain, and meanwhile, a clear plateau was disclosed in the strain range from 75 to 150%, in which the dynamic coordination bonds underwent the fracture process as explained earlier. This verified that, in the first reinforcing stage of L40Z12@300% (strain range from 75 to 150%), the relative crystallinity had little change, the rapid growth in modulus was mainly resulted from the dynamic fracture of coordination bonds. The increase trend in the relative crystallinity of L40Z12@300% was obviously accelerated as the strain reached over 150%, verifying that the SIC course followed the fracture of coordination bonds in the second-stage enhancement (strain >150%).

For comparison, the XRD patterns were also examined for the control sample L40Z12@0% without mechanical training. As shown in Fig. 3e, either before stretching (0% strain) or under the same strain, the crystalline diffraction intensity of L40Z12@0%

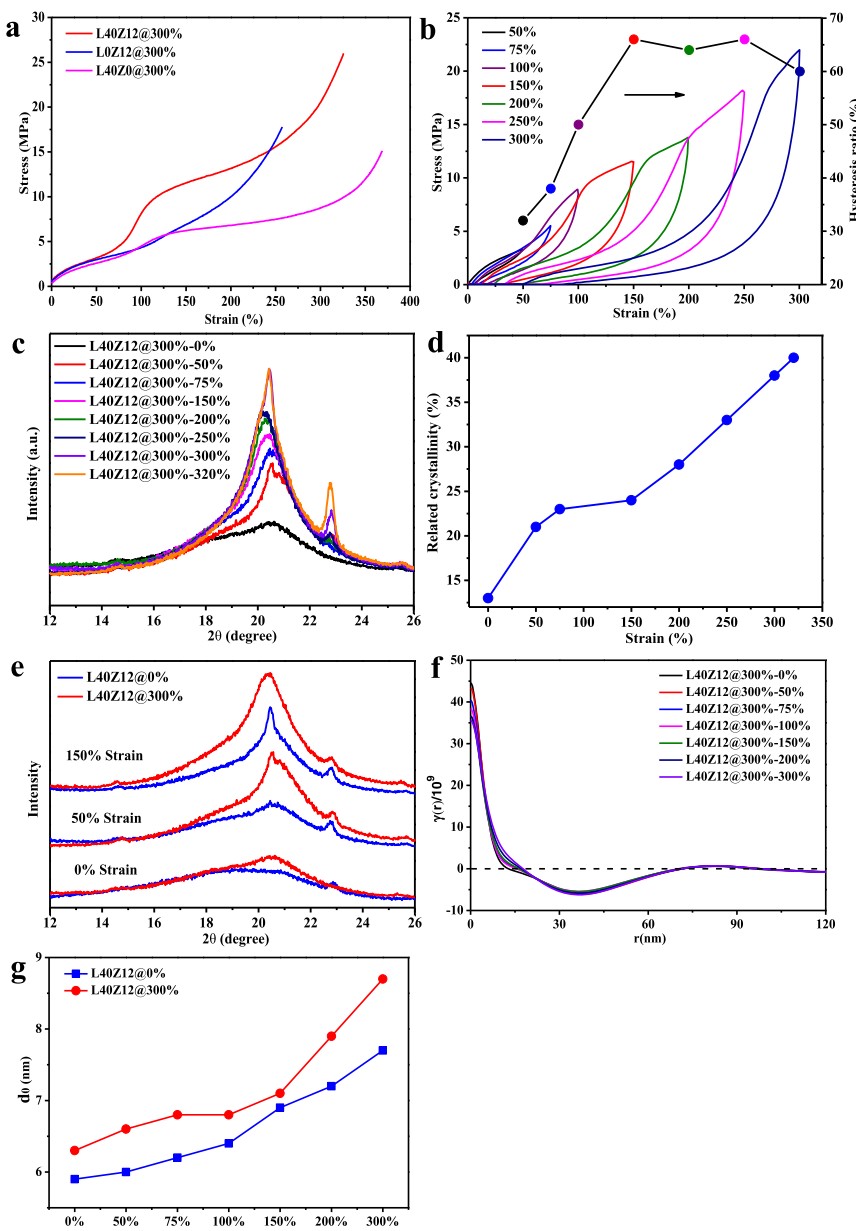

**Fig. 3 Characterizations for the self-strengthening mechanism. a** The engineering stress–strain curves of L40Z12@300%, L0Z12@300%, and L40Z0@300%. **b** The hysteresis tensile curves and the corresponding hysteresis ratio of L40Z12@300% at various strains from 50 to 300%. **c** The XRD patterns of L40Z12@300% at various fixed strains (a.u. means absorbance unit). **d** The relative crystallinity calculated from the XRD patterns of L40Z12@300%. **e** The XRD patterns of L40Z12@0% and L40Z12@300% at various fixed strains. **f** Normalized 1D correlation function curves for L40Z12@300% at various strains. **g** The core crystalline layer length ($d_0$) at different strains for L40Z12@0% and L40Z12@300%. Source Data for Fig. 3a–g are available as an associated article file.

was weaker than that of L40Z12@300%, and this difference was more obvious when the stretching strain increased. These results indicated that, in the presence of coordination bonds, the mechanical training process facilitated chain orientation in the elastomer composite and promote more obvious SIC phenomena during stretching, thus causing the strain-induced stiffening in the second-stage enhancement of L40Z12@300%.

The evolution of polymer matrix during deformation could be further revealed by in-situ tensile small-angle X-ray scattering (SAXS) analysis. As shown in Fig. 3f and Supplementary Fig. 13a–c, through analysis of the normalized one-dimensional correlation function curves of the SAXS profiles at different

strains[27], the structure parameters such as the long period length ($L$), the transition layer length ($d_{tr}$), the length of the crystalline layer plus transition layer ($l_c$), were achieved in Supplementary Table 1. The core-crystalline layer length ($d_0 = l_c - d_{tr}$) was then calculated accordingly. The evolution of these structure parameters reflected the deformation mechanism of L40Z12@300%. The core-crystalline layer length ($d_0$) increased as the strain enlarged, implying that new thicker micellar crystals formed upon stretching (Fig. 3g). Specifically, for L40Z12@300%, $d_0$ maintained constant value between 75–100% strain, consistent with the plateau revealed in the relative crystallinity analysis in Fig. 3d. After the plateau, the $d_0$ value of L40Z12@300% increased much

faster, especially when the strain was larger than 150%. This reconfirmed that, the rapid modulus growth in the first-stage enhancement of L40Z12@300% was mainly resulted from the dynamic fracture of coordination bonds, after which SIC occurred and contributed to the second-stage enhancement. However, the $d_0$ value of L40Z12@0% and L40Z0@300% increased steadily during the whole stretching course without the similar plateau (Fig. 3g and Supplementary Fig. 13d), showing typically uniform deformation of elastomers. As shown in Supplementary Table 1, comparison on the $d_0$ and $\Delta d_0$ values of L40Z12@0% and L40Z0@300% with L40Z12@300% further demonstrated that, after mechanical training, the sacrificial coordination bonds in the elastomer composite could not only stabilize chain orientation, but also promote the orientation of chain segments during the stretching course.

Based on the analysis above, the self-strengthening mechanism by mechanical training for the EPDM-based elastomer composite was proposed in Fig. 4. Firstly, through repetitive pre-stretching of mechanical training process, a certain part of chain orientation was stabilized by the coordination bonds along the pre-stretching direction (State 1 in Fig. 4), forming a more well-defined strand configuration, which would benefit for the stress transfer from elastomer matrix to the coordination bonds. Then, as the elastomer sample after mechanical training was stretched, the tensile stress–strain curve could be divided into three levels (Fig. 4). Take the sample L40Z12@300% as an example, in Level 1, the physically entangled elastic network was deformed first at small strain, during which the elastomer chain segments were tensioned (State 2 in Fig. 4). Upon further stretching to the medium strain (75–150% for L40Z12@300%), the stress was transferred to the coordination bonds, the sacrificial coordination bonds started to resist the applied stress and dynamically ruptured to dissipate the energy, which led to the substantial increase in the modulus in Level 2. Once the coordination bonds were sacrificed, the elastomer chain segments slipped from the

surface of lignin particles (State 3 in Fig. 4), which led to the decrease in the modulus (110–150%) and resulted in the S-shape turning of the stress–strain curve in Level 2. Following Level 2, further stretching to a large strain in Level 3 led to the ordered chain alignment and SIC, which offered another dramatic increment in the modulus in Level 3. Specially, the sacrificial coordination bonds after mechanical training promoted SIC more efficiently as they dynamically ruptured in a focused strain region in Level 2 and facilitated the chain alignment in Level 3. It should be noted that, without lignin, there would be no slippage of elastomer chain segments on the particle surface and the phenomenon of dual-stage enhancement would disappear, while without coordination bonds, the self-strengthening effect and the strain-adaptive stiffening would disappear, which both would result in poor mechanical performance of the elastomer composite.

In short, the sacrificial coordination bonds underwent destruction and reconstruction process in the elastomer composite, which not only stabilized the chain orientation along the pre-stretching direction after mechanical training, but also efficiently dissipated energy and promoted the oriented crystallization, thus endowing the material with strong self-strengthening and strain-adaptive stiffening performance.

In order to demonstrate the variation of related network inside the elastomer composite during the deformation course, a one-dimensional constitutive model was employed for describing the stress–strain relationship of materials. According to the characteristics of three-level stress response illustrated in Fig. 4, we assume that the molecular chain network in this elastomer composite was composed of three sub-networks, including the covalent-bond network, physically entangled network, and coordination-bond network (Fig. 5a). The detailed explanation for the one-dimensional constitutive model was provided in the Supporting Information. The value of $\sigma_0$, $\sigma_1$, and $\sigma_2$ corresponds to the stress of covalent-bond network, physically entangled

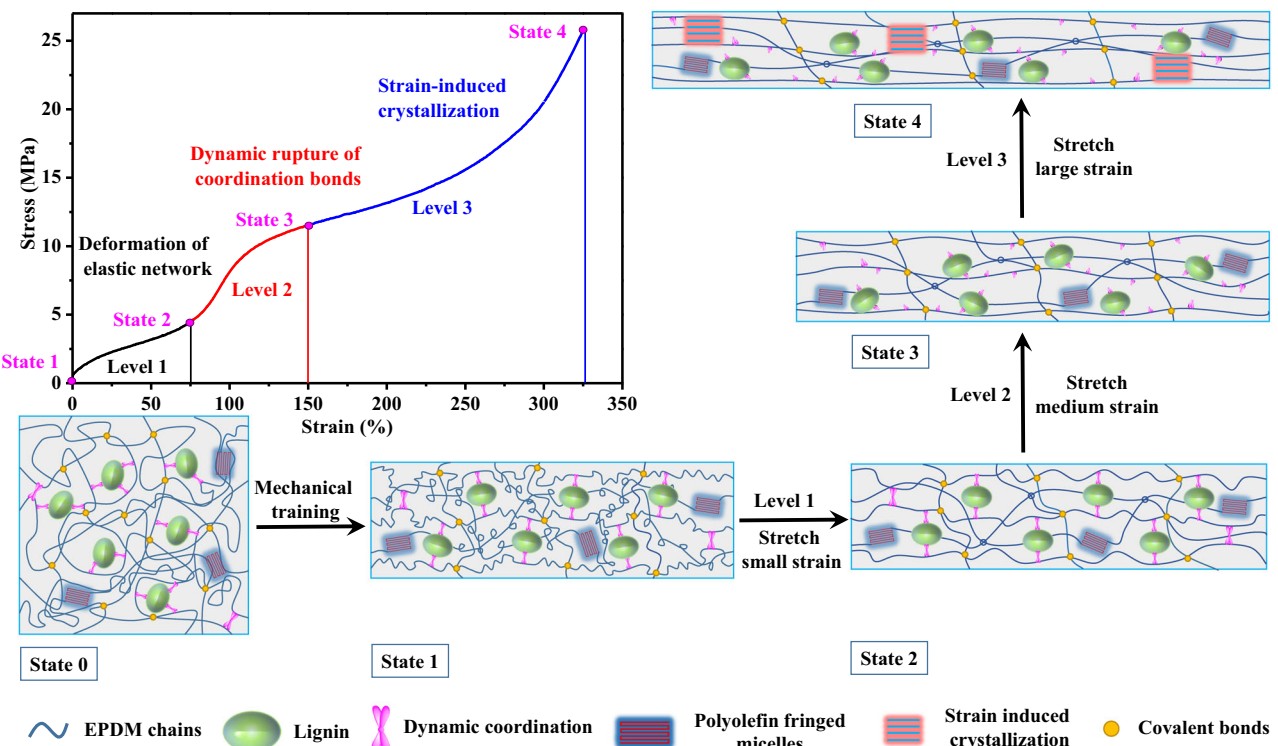

**Fig. 4 Schematic illustration.** Self-strengthening mechanism by mechanical training for the EPDM-based elastomer composite.

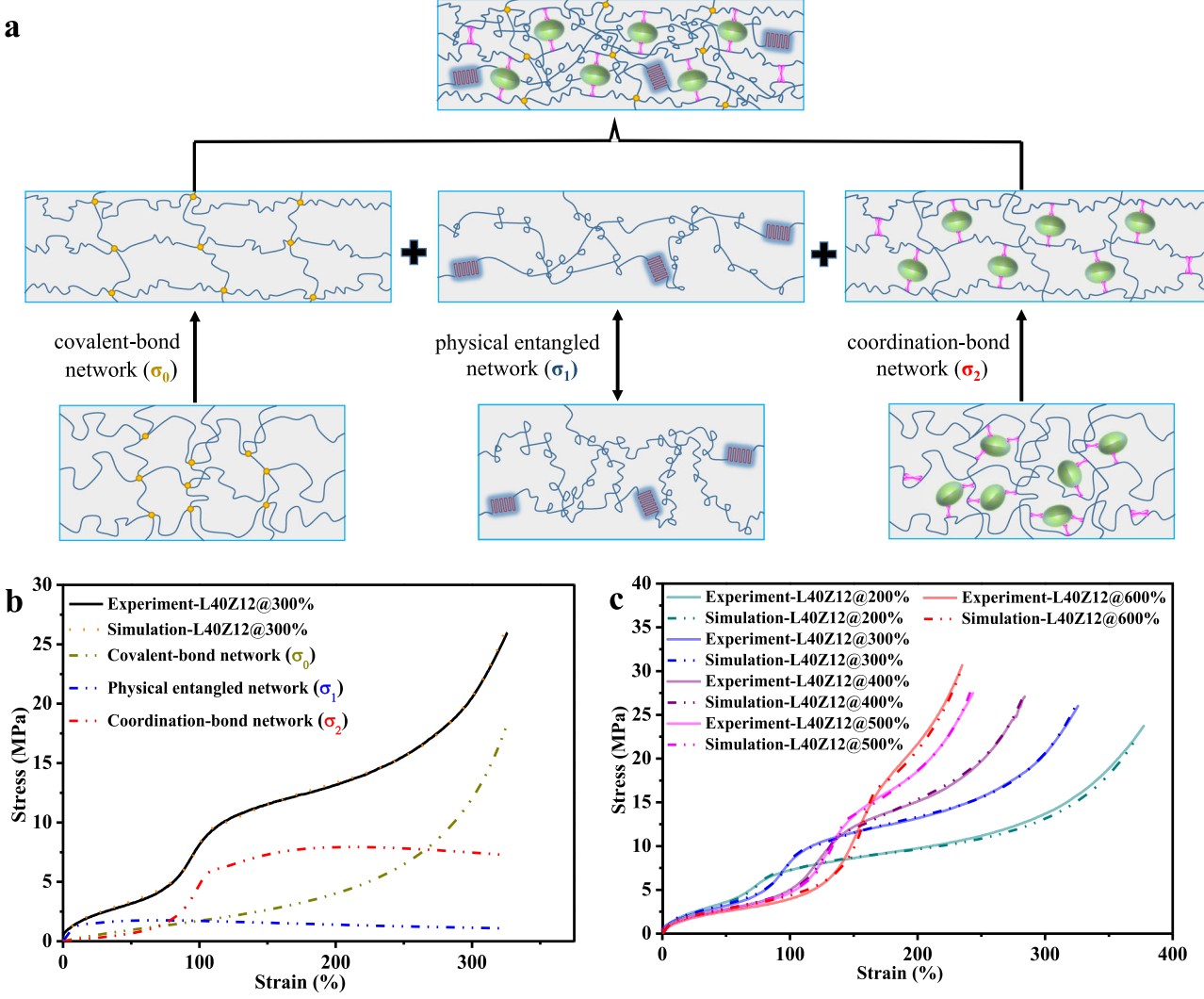

**Fig. 5 Theoretical simulation for the artificial muscle material. a** Schematic illustration of the variation of the covalent-bond network ($\sigma_0$), physical entangled network ($\sigma_1$), and coordination-bond network ($\sigma_2$) when the training strain increased. **b** The simulation of engineering stress–strain curve for L40Z12@300%, including covalent-bond network ($\sigma_0$), elastic network ($\sigma_1$), and coordination-bond network ($\sigma_2$). **c** The simulation results of engineering stress–strain curves for the samples trained at different strains. Source Data for Fig. 5b–c are available as an associated article file.

network, and dynamic coordination-bond network, respectively. As shown in Fig. 5b, c, the engineering stress–strain curves of the elastomer composite after mechanical training at different training strain were perfectly simulated by the one-dimensional constitutive model, demonstrating that the three sub-network model could accurately describe the stress–strain behaviors of the elastomer composite after mechanical training. As summarized in Supplementary Table 2, through analysis on the related structure parameters fitted from the model simulation (Supplementary Fig. 14), the mechanical property of the artificial muscle material could be facilely customized by precisely programming the input mechanical training strain, the density of coordination-bond network and covalent-bond network. Note that one merit of this strategy is that it does not require complex molecular design or sophisticated synthetic process comparing with those previously reported artificial muscle materials.

**Programmable actuation performance of the EPDM-based artificial muscle material.** The key function of skeletal muscles is the intelligent actuation performance in response to external stimulus, which is the long-term pursuit of artificial muscle

materials. As more chain alignments were stabilized along the prestretching direction under larger training strain (Supplementary Fig. 7c), L40Z12@600% was selected to study the thermal-induced actuation performance by the dynamic mechanical analyzer (DMA) under the isoforce mode. Using 1.3 MPa as a constant stress of preloading, the strain variation of L40Z12@600% against temperature was plotted in Fig. 6a. Upon heating, a substantial strain change from 21 to −20% was presented as the temperature increased from −30 °C to 90 °C, while the strain value recovered to original length again along with the temperature decreased to −30 °C on the cooling process. As the heating–cooling cycle repeated, the reversible strain variation exhibited good repeatability. The mechanism for the thermal-induced reversible strain variation was attributed to the oriented crystallization and crystallization-induced elongation (CIE) upon cooling and melting-induced contraction (MIC) upon heating of the elastomer matrix[28,29], as illustrated in Fig. 6b. The reversible actuation strain reached 41% in this work, which was the largest value ever achieved in the polyolefin-based actuation materials[30–33]. The value of actuation strain (41%) performed by the EPDM-based artificial muscle material also met the requirement of human skeletal muscle (40%). In contrast,

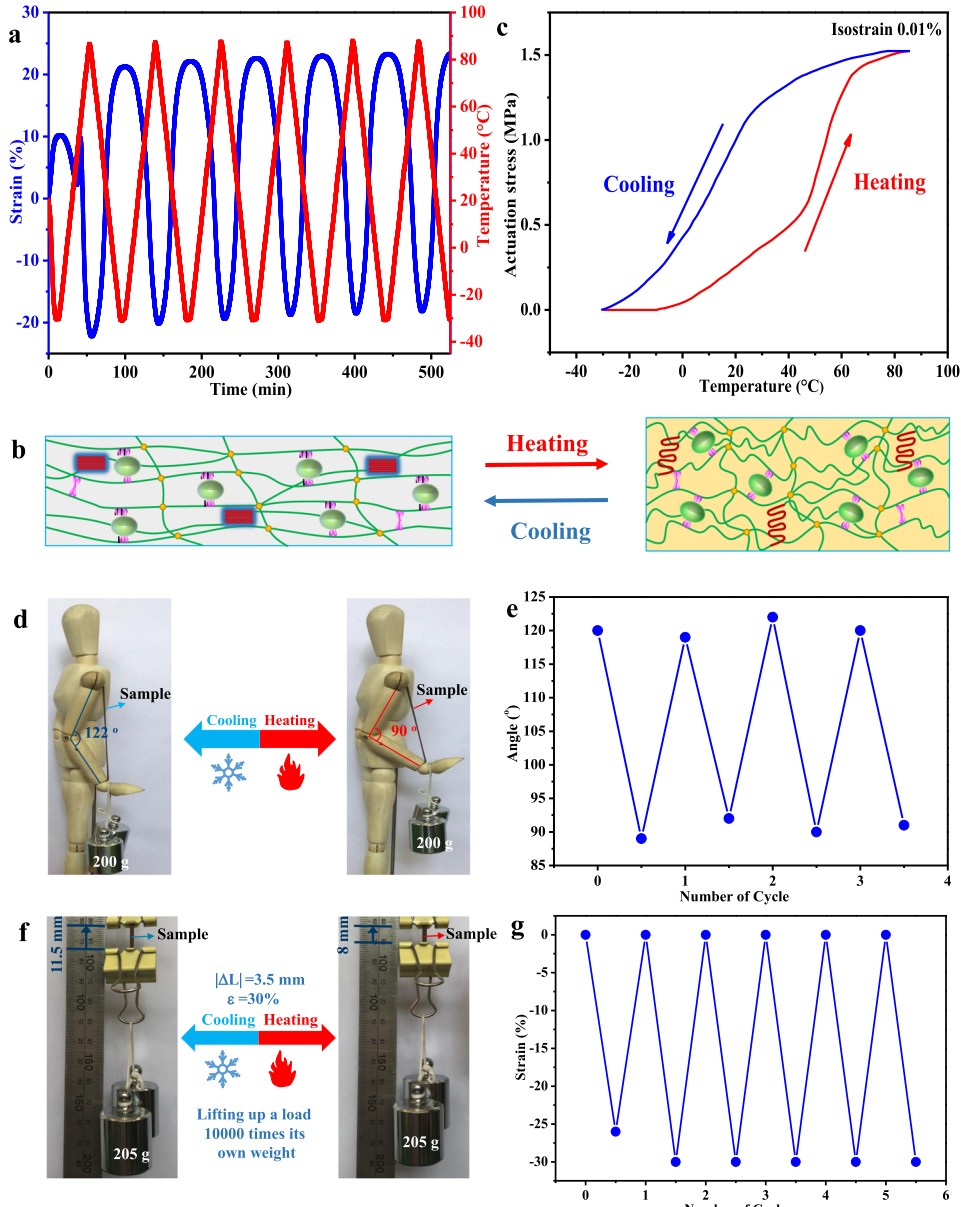

**Fig. 6 Thermal-induced actuation performance. a** Strain variation in isoforce mode and the corresponding temperature of the sample L40Z12@600% plotted against time. **b** Schematic graph of reversible actuation effect of EPDM composites after mechanical training. **c** Stress variation in isostrain mode of L40Z12@600% plotted against temperature. **d** Photographs of the angle change and **e** plot of the angle change vs cycles for the thermal actuation of L40Z12@600% lifting up a 200 g load (fire represents heating and ice represents cooling). **f** Photographs of the length change and **g** plot of the strain change vs cycles for the thermal actuation of L40Z12@600% (20 mg) bearing a load of 205 g which was 10,000 times higher than its own weight (fire represents heating and ice represents cooling). Source Data for Fig. 6a, c e, g. are available as an associated article file.

L40Z12@0% without mechanical training process exhibited only less than 25% unstable reversible strain variations upon the heating–cooling cycles (Supplementary Fig. 15a), demonstrating that the mechanical training process was an essential and effective method for improving the actuation strain of the artificial muscle material.

The actuation stress of L40Z12@600% was then measured by DMA under the isostrain mode at 0.01% strain in tension. As shown in Fig. 6c, the actuation stress generated from the contraction of the sample upon heating slowly increased as the temperature grew from −10 °C to 50 °C. Then the incremental process was accelerated when the sample was mostly melted after 50 °C (the DSC curves of the elastomer composites were provided in Supplementary Fig. 15b. In a heating–cooling cycle of −10 °C

to 90 °C, the maximum actuation stress of L40Z12@600% reached 1.5 MPa at 90 °C, demonstrating excellent ability to do work. Remarkably, the actuation stress of L40Z12@600% (1.5 MPa, 0.1% isostrain) was larger than most of previous LCE-based actuators[34–36], and was more than 4 times higher than that of the human skeletal muscle (0.35 MPa). On the contrary, the typical "thermal expansion and cold shrinkage" effect occurred in the sample L40Z12@0% during the heating–cooling cycle at isostrain mode (Supplementary Fig. 15c), verifying that the elastomer composite without the mechanical training process did not exhibit the ability to do work. The control samples L40Z0@300% without ZDMA and L0Z12@300% without lignin only generated the actuation stress of 1.0 MPa (Supplementary Fig. 15d, e). Apparently, incorporating lignin and sacrificial coordination

bonds into the elastomer network not only enhanced the mechanical properties, but also improved the actuation stress.

In order to examine the potential application of the EPDM-based artificial muscle material, the sample L40Z12@600% was attached to the arm of a puppet lifting a 200 g load as a simulation of human skeletal muscle, as shown in Fig. 6d. During the multiple heating−cooling cycles, a reversible switching angle of about 30° between the forearm and the humerus was observed (Fig. 6e and Supplementary Movie 1). When the sample of 20 mg was directly attached to the total weight of 205 g (Fig. 6f), it could repetitively lift it up and down with the actuation strain larger than 30%, which was more than 10,000 times of its own weight, as illustrated in Fig. 6f, g and Supplementary Movie 2.

As is well known, the nervous system programs the movements based on the sensory information, and then sends these instructions to the muscles in the form of electrical signals, followed by the muscles' contraction or relaxation according to the variation of potential/electricity[5]. This interesting behavior of real muscles inspired us to develop verisimilar artificial muscle

material triggered by electrical signal. For this purpose, an electric-programmable EPDM-based artificial muscle material was fabricated by substituting half mass of lignin with conductive carbon black. The prepared sample L20C20Z12@300% also exhibited the self-strengthening effect and the strain-adaptive stiffening behavior, which were similar as L40Z12@600% and could mimic the mechanical performance of skeletal muscles (Supplementary Fig. 16).

The sample L20C20Z12@300% was assembled into a program-controlled circuit, as shown in Fig. 7a. By adjusting the current value from 5 mA to 30 mA, the EPDM-based artificial muscle material was able to lift up a 505 g load (2000 times its own weight), with the reversible motion distance ($|\Delta L|$) of 7 mm (13% strain), and this behavior was repeatable (Fig. 7b and Supplementary Movie 3). When the current value was adjusted between 0 mA and 25 mA, the reversible actuation strain could be further increased to 20% (Supplementary Fig. 17). As shown in Fig. 7c, both the actuation strain and the sample temperature increased with the current value. The elastomer composite converted the

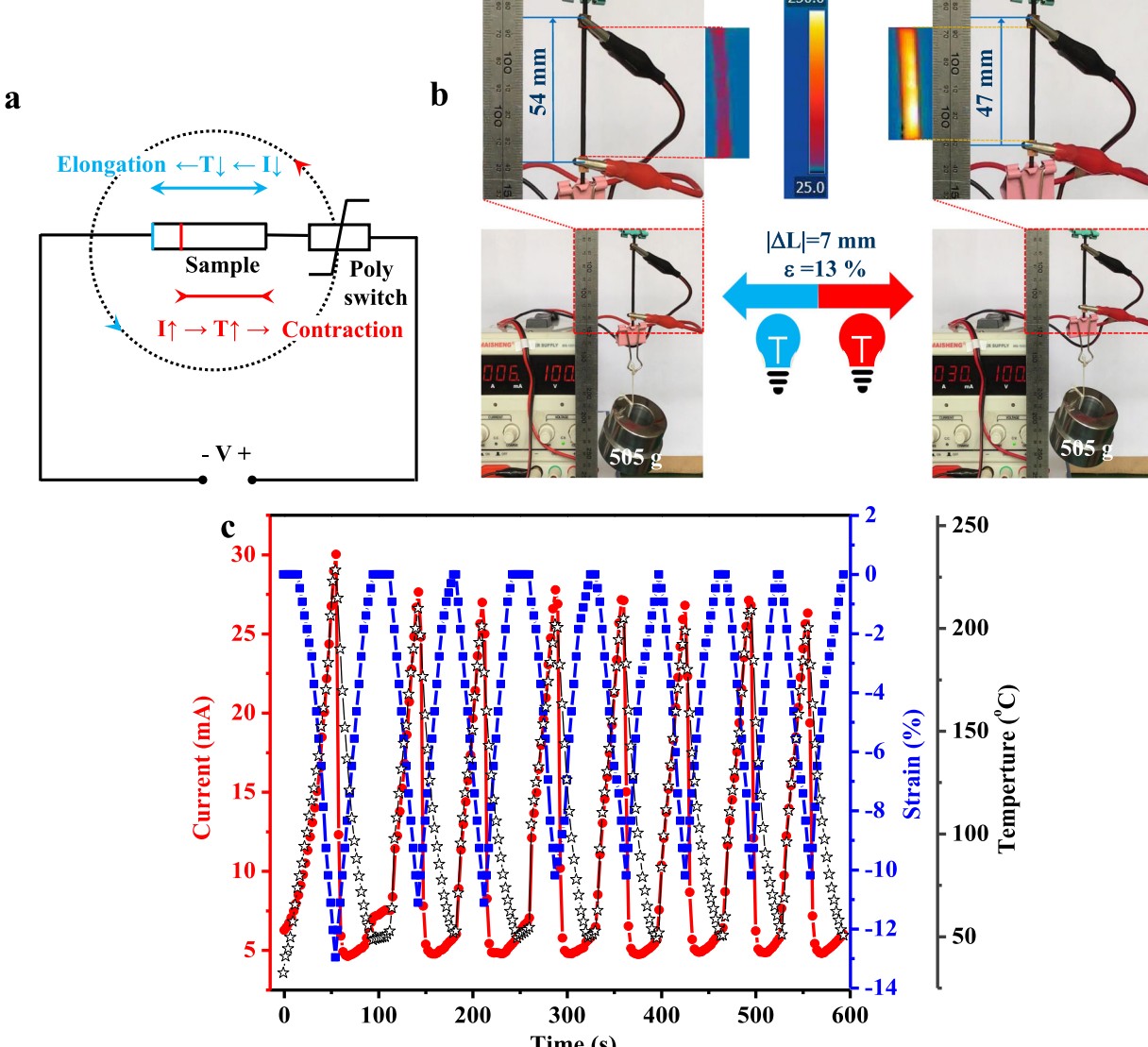

**Fig. 7 Electric-programmable actuation performance. a** Schematic diagram of the circuit for the electric-programmable actuation tests. **b** Photographs of the length change of L20C20Z12@300% under varied current signals (the color change of bulbs represent the variation of the current). **c** Quantitative evaluation of the variation of actuation strain in response to the current signal in the electric-triggered actuation tests. Source Data for Fig. 7c is available as an associated article file.

thermal energy into macroscopic actuations in accordance to the current signals, which is a good bionic duplicate of skeletal muscles converting the chemical energy source (ATP) into macroscopic motions according to the current signals sent by the nervous system. Note that the current values (5–30 mA) adopted for the actuation control in this work was quite close to fit for the demand of industrial standard current-control signal (4–20 mA). The EPDM-based artificial muscle material with electric-programmable actuation performance has demonstrated a great potential for the practical industrial application.

## Discussion

In conclusion, inspired by the dynamic sacrificial bonds in bio-materials and the self-strengthening mechanism of skeletal muscles by physical exercise, we have established a facile strategy for preparing the programmable artificial muscle material by integrating the superiority of sacrificial coordination bonds and mechanical training process. The biomass lignin was introduced into the commercially available EPDM as natural green reinforcer to serve as ligands for the construction of coordination bonds, followed by a repetitive mechanical training process. An obvious dual-stage enhancement in modulus was obtained for the elas-tomer composite after the mechanical training process. It was demonstrated that the destruction and reconstruction process of the sacrificial coordination bonds along the pre-stretching direction of repetitive training helped to form a more well-defined strand configuration, which not only contributed efficient energy dissipation in the first-stage enhancement, but also pro-moted the strain-induced crystallization for the second-stage enhancement, thus endowing the material with strong self-strengthening by mechanical training and strain-adaptive stif-fening performance, in which the elastic modulus could increase 16.5 times from 2 MPa to 33.2 MPa as the strain increased. The simulation results of one-dimensional constitutive model demonstrated that the mechanical property of the EPDM-based artificial muscle material could be customized by precisely pro-gramming the mechanical training strain, the density of coordination-bond network and covalent-bond network. The EPDM-based artificial muscle material could actuate program-mable reciprocating motions in accordance to the stimulation of heat and electricity, with the actuation stress and actuation strain reached 1.5 MPa and 41%, respectively. Most importantly, this EPDM-based elastomer composite is the first commercial polyolefin-derived artificial muscle material that fulfills all basic requirements for macroscopic mechanical performance of skeletal muscles including the actuation strain (>40%), actuation stress (>0.35 MPa), characteristics of self-strengthening by mechanical training, strain-adaptive stiffening with more than 10 times increase in modulus, and heat/electric programmable actuation. We hope this facile strategy via combing the sacrificial coordi-nation bonds and mechanical training process to be expanded in the fabrication of sophisticated intelligent materials using easily available green raw materials by means of easily industrial scale-up. It not only provides a method for the intellectualization of traditional elastomer materials, but also provides a direction for the high-value utilization of industrial biomass resources such as lignin.

## Methods

**Materials**. EPDM 3745 P (ethylene 70 wt%, propylene 29.5 wt%, ethylidene nor-bornene 0.5 wt%) with antioxidant 1010 was obtained from Dow Chemical. Lignin was provided by Shanghai Dongsheng Co., Ltd. Zinc dimethacrylate (ZDMA) was purchased from Sigma-Aldrich. Conductive carbon black (Ketjen Black, ECP600JD, Japanese LION) was purchased from Suzhou Yilongsheng Energy Technology Co., Ltd. Other ingredients such as zinc oxide (ZnO), stearic acid (SA), bis(tert-butyldioxyisopropyl) benzene (BIPB), and triallyl isocyanurate (TAIC) are

industrial products and used as received. Poly-switch (PPTC) was provided by Shenzhen Tairuiyuan Technology Co., Ltd.

**Fabrication methods**. Firstly, EPDM (100 phr) was mixed with lignin (40 phr) at 50 rpm and 80 °C for 8 min in an internal mixer (Guangdong Lina Co., Ltd., Dongguan, China). Then, a certain amount of ZDMA was added and mixed for 10 min. Next, all other ingredients such as ZnO (5 phr), SA (1 phr), BIPB (1 phr), TAIC (0.5 phr), were added and mixed for another 5 min, followed by hot-pressing at 170 °C for 20 min. Finally, the vulcanized elastomer was repetitively stretched for 250–1000 times at a fixed strain by electronic universal testing machine (MTS, China) to get the artificial muscle materials. All the artificial muscle materials were rested for half an hour prior to use, and were then cut into the needed shape (dumbbell or ribbon) for further study. The sample was named as LxZy@m%, with "L" standing for lignin, "Z" for ZDMA, and the numbers x and y for the parts per hundred of EPDM (100 phr), "@" for mechanical training, and the number m for the mechanical training strain. For instance, L40Z12@300% means lignin 40 phr, ZDMA 12 phr, and the fixed 300% training strain. L40Z12@0% means the sample was not submitted for mechanical training. The "C" in L20C20Z12@300% repre-sents for conductive carbon black. The sample L40Z0@300% without ZDMA and the sample L0Z12@300% without lignin were compared by mechanical training at 300% strain for 250 cycles.

**Characterization**. Bruker Vertex 70 FTIR spectrometer (Bruker, Germany) was used to collect Fourier transform infrared (FTIR) spectra at the attenuated total reflection (ATR) mode. The sample was stretched to a fixed strain (0%, 75%, 150%) before testing, and the results were normalized at 2918 cm$^{-1}$ to evaluate the var-iation of coordination bonds.

The scanning electron microscopy (SEM) was conducted on a Hitachi UHR FE-SEM SU8220 instrument (Hitachi, Tokyo, Japan) with an accelerating voltage of 5 kV, and the samples were sputtered with a thin gold film before test to enhance the conductivity.

Mechanical training and tensile tests were carried out on a CMT electronic universal testing machine (MTS, China) with the speed of 200 mm/min at room temperature. The loss energy/tensile toughness in the hysteresis tensile curves were calculated from the integral area under the loading and unloading curves in the cyclic tensile tests.

Dynamic mechanical analyzer (DMA) was used to measure the actuation stress and actuation strain of the samples. In the isoforce tests, constant stress (ranging from 1.0 MPa to 1.3 MPa) was first applied to the sample, the sample was cooled to −30 °C, then the actuation strain was recorded along with the temperature varied from −30 °C to 90 °C at a rate of 3.0 °C/min. The actuation strain was defined as $(L - L_0)/L_0$, where $L$ is the real-time length of the sample measured at any temperature and $L_0$ is the initial length before testing. In the isostrain experiments, constant strain was first fixed at 0.01%, the sample was cooled to −30 °C, then the actuation stress was recorded along with the temperature ranged from −30 °C to 90 °C at a rate of 3.0 °C/min. The range of actuation stress in the isostrain tests was between 0.9 MPa and 1.6 MPa.

X-ray diffraction (XRD) test was performed on a X-ray diffractometer (PANalytical, X'pert3 Powder) using Cu radiation with incident wavelength of 0.1541 nm. The sample was stretched to the fixed strain before testing, and was then scanned at room temperature with the scanning step of 0.013° and the scanning speed of 15 s/step. The resulting profiles were transformed to the relative crystallinity by the software (Jade 6).

Small-angle X-ray scattering (SAXS) experiments were conducted by a Bruker Nanostar U instrument. The wavelength of X-ray source was 1.54 Å and the detector was Vantec 2000 with the sample-to-detector distance of 1078 mm. The exposure time was set 600 s for each scan at each strain.

In the thermo-stimulus actuation test, the sample was heated by a hair drier and cooled down, the weights were lifted up and down, a switching angle and a reversible strain were observed and recorded.

In the electric-programmable actuation test, the sample L20C20Z12@300% was assembled into a program-controlled circuit, as shown in Fig. 7a. According to Joule's law, under a constant voltage of 100 V, the current passing through the sample generated heat, thus increased the temperature inside the sample, leading to the sample contraction. When the current was turned down or decreased, the sample cooled down and relaxed to its original length. The temperature inside the sample was detected by the infrared camera (FLIR T530, 30 mK thermal sensitivity). The current was output from a DC power supply (MAISHENG, MS-1001D).

## Data availability

The source data that support the findings of this study are available in materialscloud with the identifier https://doi.org/10.24435/materialscloud:9a-7y[37]. Source data are provided with this paper.

## Code availability

All the code that support the findings of this study are available from the corresponding author on reasonable request.

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

## Acknowledgements

The authors gratefully thank the National Natural Science Foundation of China (22038004, 22078116, 21706082), Guangdong Provincial Key Research and Development Program (2020B1111380002), Natural Science Foundation of Guangdong Province (2019A1515012154, 2018B030311052) for the financial supports.

## Author contributions

Z.T. and W.L. designed and performed the experiments and wrote the manuscript. J.W. performed the theoretical simulations. J.H. participated in data characterization for SAXS. J.L. took part in data recording for actuation experiment. And H.L. participated in the data discussion for XRD. W.L. and X.Q. revised the manuscript and supervised the research.

## Competing interests

The authors declare no competing interests.
