## [Peer Review File · Nature Communications]

REVIEWER COMMENTS

Reviewer #1 (Remarks to the Author):

The manuscript claims a new artificial muscle formulation that achieves: (i) self-strengthening by rearrangement of sacrificial bonds, and (ii) strain-adaptive stiffening that (iii) can achieve high actuation stress / strain and (iv) can be triggered with both heat and electric current. Of these claims, the third one possibly warrants publication in Nature Communications and should be appropriately emphasized, keeping the manuscript more concise. While the scientific explanation on how (iii) is achieved is important to discuss, its physical basis is not novel. Indeed, strand reconfiguration within reversibly binding networks under mechanical strain is known and quite well understood. Self-strengthening when exposed to stress, toughening, and self-healing are frequently reported for covalently adaptable networks, and for networks with reversible ionic bonds like the Zn(2+)-carboxylate interaction described here. The stiffening of material when loaded with more ZMDA is obvious, and the connection between repetitive training and a more well defined stress-strain curves is interesting, but not really surprising or novel. The topic of strain-hardening due to crystallization is decades old; however the use of reversible bonds to prevent failure-- allowing for strain-induced crystallization to kick-in-- is more interesting.

1. The experimental effort (FTIR, control samples) to establish the role of coordination bonds, lignin, and repetitive mechanical training on stress-strain behavior is commendable, though the discussion may be summarized with details moved to the SI.
2. Reporting that the material can lift more than 10^4 times itself in the abstract is meaningless unless a strain range is included. E.g. weight hanging on any actuator could be moved a small strain.
3. Dynamic bonds were found to fracture between 75% and 150%. If this is the case, and if applied stress is released at 150%, does the resulting specimen exhibit a different stress-strain curve?
4. On a related note: shouldn't State 1 in Figure 4 relax with time in a rubbery state at room temperature? Without mechanical stress, what maintains the configurational bias? What is the rate of strand relaxation?
5. It would be good to comment on the tensile strength at failure; how close are experiments to the ultimate tensile strength?
6. In my opinion, the discussion of the 1D constitutive model could be moved to SI; however, the Editor may have a different idea on how to proceed.
7. The large actuation strain range is an exciting result. The nature of this type of actuation originates from configurationally biased coils within an elastic network and has been previously described (ACS Macro Letters, v4, p115) and should be appropriately referenced.
8. The actuation experiments are all conducted at relatively low strain-- presumably not to exceed stresses that may rupture reversible bonds. Comments should be added on the actuation operation strain and stress range.
9. The stimulus provided by electrical means is icing on the cake, but again, it is not novel and may be de-emphasized somewhat.

Other issues

10. It would be good to clearly define how the samples are named up-front.
11. There is no carboxylate group in EPDM, as described on line 126 on page 6. I suspect the authors were referring to ZDMA.

12. The stress-strain curve for LOZ12@300% should be displayed somewhere to confirm the disappearance of dual-stage enhancement.

13. Page 11 and page 13: "plateau" instead of "platform".

Reviewer #2 (Remarks to the Author):

Biomimetic High-Performance Artificial Muscle Built on Sacrificial Coordination Network and Mechanical Training Process

In this manuscript, the authors have described the development of a biomimetic polymer network that consists of permanent covalent bonds and sacrificial coordination complexes. The authors have proposed a mechanical strengthening regime, where only the sacrificial bonds break during the training process and the reformed bonds are stronger than those in the original network. The SIC has been extensively characterized and equated to rebuilding and reconstruction of muscles that undergo repeated strenuous exercise. Other smart responses to stimuli were also evaluated. While the conception of the material design and characterization methods deserve merit, I have some suggestions and questions that need to be addressed prior to publication:

- Figure 1 c and d missing from the collage of images in Figure 1?
- The paper below has used similar materials and reaction mechanism but for some reason has been omitted from the references cited by the authors of the manuscript that is under review. <https://www.sciencedirect.com/science/article/pii/S0142941812000955>; Y Chen, 2012, Cited by 64
The authors must distinguish their approach from this previously published paper. It is essential to point out the significance of their work for consideration in this journal.
- Page 5, Line 112 "During the vulcanization process, the unsaturated double bonds in EPDM reacted with the methacrylate groups in ZDMA, generating the ZDMA modified EPDM with functional carboxylate groups (Fig. 1b). The Zn-based coordination bonds could form between the carboxylate groups grafted on EPDM backbones and the polar functional groups in lignin such as the phenoxy and carboxylate groups..."
- Figure 1b is referenced here. However, Figure 1b only shows the independent monomers and does not show the reaction mechanism. It is recommended that the authors show the reaction mechanism for clarity.
- Figure 1 represents a bivalent metal ion (Zn^{2+}) forming a tris-complex. Here are my questions and suggestions related to that –
- What electronic configuration supports formation of a tris-complex with a bivalent ion?
- The chemical structure of lignin can be represented in several ways, sometimes depending on the source of origin. It appears that the authors received this lignin as a gift. It is suggested that the structure be shown as opposed to a green filled circle with several –OH and –COOH groups.
- Some commonly available materials at Sigma Aldrich <https://www.sigmaaldrich.com/materials-science/material-science-products.html?TablePage=20204096> show the presence of methoxy, H, and –OH on the benzene ring. If the authors' lignin looks anything like anyone of these products, then the schematic in 1b showing the tris-complex inside the pink colored circle would be inaccurate.

It is critical that we understand the chemical structure of the polymer before drawing any conclusions related to its enhanced mechanical performance.

- Currently, the peaks in the reference paper (#23) do not align very well with the ones observed

here by the authors. Here's another reference for FTIR data -

- Boeriu, C. G., Bravo, D., Gosselink, R. J., & van Dam, J. E. (2004). Characterisation of structure-dependent functional properties of lignin with infrared spectroscopy. *Industrial crops and products*, 20(2), 205-218.

- In Figures 6d and 6f, it might be worthwhile to indicate where the L40Z12@600% sample actually is. While it is pretty evident that the sample is between the two binder clips in case of 6f, the same cannot be said for 6d.

- There are several sentences in the '2.3 Programmable actuation performance of the EPDM-based artificial muscle material' section that should also be a part of the methods section. For instance- "As the material was heated by a hair drier and cooled down, the weights were lifted up and down, a switching angle of about 30o between the forearm and the humerus was observed."

- Up until the programmable actuation section, the authors have examined L40Z12@300% in great detail. It is essential for the authors to mention the rationale behind switching to L40Z12@600% in the actuation section. The authors also switched back to L20C20Z12@300% in the electric programming and actuation section without much reasoning behind choosing this specific composition.

- Page 22 - "The self-strengthening effect and the strain-adaptive stiffening behavior of L20C20Z12@300% were similar as L40Z12@600% and could mimic the mechanical performance of skeletal muscles."

The authors may have forgotten to add the data supporting this claim.

Point by point reply

Reviewer #1 (Remarks to the Author):

The manuscript claims a new artificial muscle formulation that achieves: (i) self-strengthening by rearrangement of sacrificial bonds, and (ii) strain-adaptive stiffening that (iii) can achieve high actuation stress / strain and (iv) can be triggered with both heat and electric current. Of these claims, the third one possibly warrants publication in Nature Communications and should be appropriately emphasized, keeping the manuscript more concise. While the scientific explanation on how (iii) is achieved is important to discuss, its physical basis is not novel. Indeed, strand reconfiguration within reversibly binding networks under mechanical strain is known and quite well understood. Self-strengthening when exposed to stress, toughening, and self-healing are frequently reported for covalently adaptable networks, and for networks with reversible ionic bonds like the Zn^{2+} -carboxylate interaction described here. The stiffening of material when loaded with more ZMDA is obvious, and the connection between repetitive training and a more well defined stress-strain curves is interesting, but not really surprising or novel. The topic of strain-hardening due to crystallization is decades old; however the use of reversible bonds to prevent failure--allowing for strain-induced crystallization to kick-in-- is more interesting.

Response: Thank you for the kind comments. The novelty of this work is integrating the superiority of sacrificial coordination bonds and mechanical training process. The destruction and reconstruction process of the sacrificial coordination bonds along the pre-stretching direction of repetitive training help to form a more well defined strand configuration, which not only contributed efficient energy dissipation in the first-stage enhancement, but also promoted the strain-induced crystallization for the second-stage enhancement, thus endowing the material with strong self-strengthening and strain-adaptive stiffening performance, and also excellent actuation performance with high actuation stress/strain. Another highlight of this work is to provide a

direction for the high-value utilization of industrial biomass resources such as lignin.

1. The experimental effort (FTIR, control samples) to establish the role of coordination bonds, lignin, and repetitive mechanical training on stress-strain behavior is commendable, though the discussion may be summarized with details moved to the SI.

Response: Thank you for the suggestion. Now the discussion on the FTIR analysis of the role of coordination bonds, lignin, and repetitive mechanical training on stress-strain behavior is moved from previously Fig. 3c to the Supplementary Fig. 12.

2. Reporting that the material can lift more than 10^4 times itself in the abstract is meaningless unless a strain range is included. E.g. weight hanging on any actuator could be moved a small strain.

Response: Thanks. The strain range of 30% is included in the abstract.

3. Dynamic bonds were found to fracture between 75% and 150%. If this is the case, and if applied stress is released at 150%, does the resulting specimen exhibit a different stress-strain curve?

Response: Yes, it does. As shown in figure 3b, after the applied stress was released at 150%, owing to the fracture of coordination bonds, the first-stage enhancement in the strain range of 75%-150% disappeared in the immediate reloading stress-strain curve as illustrated by the green curve.

4. On a related note: shouldn't State 1 in Figure 4 relax with time in a rubbery state at room temperature? Without mechanical stress, what maintains the configurational bias? What is the rate of strand relaxation?

Response: Thanks. After mechanical training, part of rubber strands would relax, but a certain part of chain alignment was stabilized by the regenerated coordination bonds and the newly formed microcrystals after mechanical training, which was illustrated by the generally stable residual strain after mechanical training shown in Supplementary

Fig. 7 and the relative higher crystallinity after mechanical training shown in Supplementary Fig. 8. The State 1 in Figure 4 represents the state stabilized by the coordination bonds after partly relaxation.

The rate of strand relaxation was reflected by the stress relaxation curve of L40Z12@300% depicted in Supplementary Fig 9. Only part of rubber chain strands relaxed after mechanical training.

5. It would be good to comment on the tensile strength at failure; how close are experiments to the ultimate tensile strength?

Response: Thanks. The comment on the tensile strength at failure was depicted at Page 9, Line 154-156 in the Revised manuscript, “As the training strain increased from 0% to 600%, the tensile strength of the elastomer composite increased gradually from 24.8 MPa to 30.7 MPa”.

In the mechanical training process, the training strain was between 200% and 600%, the corresponding stress was between 3 MPa and 7.3 MPa. In the isoforce tests, the constant stress ranged from 1.0 MPa to 1.3 MPa. In the isostrain tests, the range of actuation stress was between 0.9 MPa and 1.6 MPa. Obviously, the experiments were far from the ultimate tensile strength.

6. In my opinion, the discussion of the 1D constitutive model could be moved to SI; however, the Editor may have a different idea on how to proceed.

Response: Thank you for the suggestion. The discussion of the 1D constitutive model was already provided in the SI, as shown in the Section of “Simulation results of the one-dimensional constitutive model” in the SI document. Here we prefer to keep the major conclusions from the model simulation in the main manuscript for better understanding the mechanism. However, if the reviewer and Editor prefer to move the major conclusions from the model simulation to SI, we can revise it then.

7. The large actuation strain range is an exciting result. The nature of this type of actuation originates from configurationally biased coils within an elastic network and

has been previously described (ACS Macro Letters, v4, p115) and should be appropriately referenced.

Response: Thank you for pointing out. This paper is now cited in reference [28].

8. The actuation experiments are all conducted at relatively low strain-- presumably not to exceed stresses that may rupture reversible bonds. Comments should be added on the actuation operation strain and stress range.

Response: Thanks. In the isoforce tests, the constant stress was ranging from 1.0 MPa to 1.3 MPa. In the isostrain tests, the range of actuation stress was between 0.9 MPa and 1.6 MPa. Thus, it is far from the tensile strength at failure and less than the stresses that may rupture reversible bonds. These comments are now added in the Characterization section in the revised manuscript.

9. The stimulus provided by electrical means is icing on the cake, but again, it is not novel and may be de-emphasized somewhat.

Response: Thank you for the suggestion. The actuation stimulated by electricity was de-emphasized in the revised manuscript.

Other issues

10. It would be good to clearly define how the samples are named up-front.

Response: Thank you for pointing out. The principle for sample naming was added in the first paragraph of Section 1.1.

11. There is no carboxylate group in EPDM, as described on line 126 on page 6. I suspect the authors were referring to ZDMA.

Response: Thanks. The mistake was corrected in the revised manuscript.

12. The stress-strain curve for L0Z12@300% should be displayed somewhere to confirm the disappearance of dual-stage enhancement.

Response: Thanks. The tensile stress-strain curve for LOZ12@300% was displayed as blue line in Fig 3a.

13. Page 11 and page 13: "plateau" instead of "platform".

Response: Thanks. "platform" was replaced by "plateau" in the revised manuscript.

Reviewer #2 (Remarks to the Author):

Biomimetic High-Performance Artificial Muscle Built on Sacrificial Coordination Network and Mechanical Training Process

In this manuscript, the authors have described the development of a biomimetic polymer network that consists of permanent covalent bonds and sacrificial coordination complexes. The authors have proposed a mechanical strengthening regime, where only the sacrificial bonds break during the training process and the reformed bonds are stronger than those in the original network. The SIC has been extensively characterized and equated to rebuilding and reconstruction of muscles that undergo repeated strenuous exercise. Other smart responses to stimuli were also evaluated. While the conception of the material design and characterization methods deserve merit, I have some suggestions and questions that need to be addressed prior to publication:

Response: Thank you for the kind comments.

1. Figure 1 c and d missing from the collage of images in Figure 1?

Response: Thank you for pointing out. It was a mistake. The missing images of Figure 1 c and d were depicted in Figure 2 c and d, and the redundant sentences for Figure 1 c and d were deleted.

2. The paper below has used similar materials and reaction mechanism but for some reason has been omitted from the references cited by the authors of the manuscript that is under review.

<https://www.sciencedirect.com/science/article/pii/S0142941812000955>; Y Chen, 2012,

Cited by 64

The authors must distinguish their approach from this previously published paper. It is essential to point out the significance of their work for consideration in this journal.

Response: Thank you for your advice. This paper was cited as reference [23] in the revised manuscript. This previously published paper reported PP/EPDM-based dynamically vulcanized thermoplastic olefin with zinc dimethacrylate. The ZDMA grafted product improved the compatibility between EPDM and PP phases. The target was totally different from our work.

In our work, we tried to use lignin as green reinforcer in elastomer. The biomass lignin is the largest aromatic biopolymer in nature. However, due to the easy aggregation of lignin in polymer matrix and the weak interfacial interactions between lignin and nonpolar elastomer, the pursuit of high-performance lignin/elastomer composite has been a long-term challenge and has attracted hot attention in both the industrial and academic community. As lignin is rich in oxygen-containing polar groups, we incorporated lignin as natural ligands for the construction of interfacial coordination bonds. Our strategy involved in reactive coordination bonds and coordinative lignin to prepare the high performance EPDM elastomer composite, providing a direction for the utilization of lignin in elastomer. Two sentences were added at the end of the first paragraph in Section 1.1 and Supplementary Fig. 4 was added in SI to point out the significance of coordinative reinforcer lignin.

3. Page 5, Line 112 “During the vulcanization process, the unsaturated double bonds in EPDM reacted with the methacrylate groups in ZDMA, generating the ZDMA modified EPDM with functional carboxylate groups (Fig. 1b). The Zn-based coordination bonds could form between the carboxylate groups grafted on EPDM backbones and the polar functional groups in lignin such as the phenoxy and

carboxylate groups...”

- Figure 1b is referenced here. However, Figure 1b only shows the independent monomers and does not show the reaction mechanism. It is recommended that the authors show the reaction mechanism for clarity.

Response: Thank you for the suggestion. The reaction mechanism was added in Supplementary Fig 3.

4. Figure 1 represents a bivalent metal ion (Zn^{2+}) forming a tris-complex. Here are my questions and suggestions related to that –

- What electronic configuration supports formation of a tris-complex with a bivalent ion?

Response: Thanks. <https://www.mdpi.com/1420-3049/25/24/5814>;

M. Porchia, et al. Zinc Complexes with Nitrogen Donor Ligands as Anticancer Agents, *Molecules* 2020, 25(24), 5814; <https://doi.org/10.3390/molecules25245814>;

We added this paper in Reference [24].

The sentence in this paper might answer your question.

“The Zn^{2+} d10 configuration, and the consequent absence of d-d transition, could be seen as a limit for the spectroscopic characterization of Zn derivatives, together with their diamagnetism and white colour, but on the other hand the absence of ligand field stabilization can guarantee highly flexible coordination geometry determined only by the charge and steric hindrance of the ligands. In biological systems zinc can be tetra-, penta-, or hexacoordinated to N, O or S donor atoms comprised in histidine, glutamate/aspartate, and cysteine residues, or to water molecules with a tetrahedral, pyramidal, or octahedral coordination geometry.”

5. The chemical structure of lignin can be represented in several ways, sometimes depending on the source of origin. It appears that the authors received this lignin as a gift. It is suggested that the structure be shown as opposed to a green filled circle with several –OH and –COOH groups.

Response: Thanks. The detailed chemical structure of lignin is now presented in

Supplementary Fig. 2.

6. Some commonly available materials at Sigma Aldrich <https://www.sigmaaldrich.com/materials-science/material-science-products.html?TablePage=20204096> show the presence of methoxy, H, and –OH on the benzene ring. If the authors' lignin looks anything like anyone of these products, then the schematic in 1b showing the tris-complex inside the pink colored circle would be inaccurate.

It is critical that we understand the chemical structure of the polymer before drawing any conclusions related to its enhanced mechanical performance.

Response: Thank you for pointing out. The lignin structure in Figure 1b was corrected, methoxy is added on the benzene ring and the detailed chemical structure of lignin is now presented in Supplementary Fig. 2.

7. Currently, the peaks in the reference paper (#23) do not align very well with the ones observed here by the authors. Here's another reference for FTIR data –

• Boeriu, C. G., Bravo, D., Gosselink, R. J., & van Dam, J. E. (2004). Characterisation of structure-dependent functional properties of lignin with infrared spectroscopy. *Industrial crops and products*, 20(2), 205-218.

Response: Thanks. The original reference paper (#23) is now replaced by this paper. This paper is cited as reference [25] in the revised manuscript.

8. In Figures 6d and 6f, it might be worthwhile to indicate where the L40Z12@600% sample actually is. While it is pretty evident that the sample is between the two binder clips in case of 6f, the same cannot be said for 6d.

Response: Thank you for the suggestion. The sample position is now marked in Figures 6d and 6f.

9. There are several sentences in the '2.3 Programmable actuation performance of the EPDM-based artificial muscle material' section that should also be a part of the

methods section. For instance- “As the material was heated by a hair drier and cooled down, the weights were lifted up and down, a switching angle of about 30° between the forearm and the humerus was observed.”

Response: Thanks. These sentences were shortened in the main text and some were moved to the Methods-Characterization section.

10. Up until the programmable actuation section, the authors have examined L40Z12@300% in great detail. It is essential for the authors to mention the rationale behind switching to L40Z12@600% in the actuation section. The authors also switched back to L20C20Z12@300% in the electric programming and actuation section without much reasoning behind choosing this specific composition.

Response: Thanks. As mentioned previously in the manuscript and Supplementary Fig.7, more dynamic coordination bonds were involved in destruction and reconstruction course under larger mechanical training strain, and thus more chain alignments were stabilized along the prestretching direction under larger training strain. As chain alignment is closely related to the actuation performance, L40Z12@600% was selected rather than L40Z12@300% to get the better thermal-induced actuation performance. Now one sentence was added in the first paragraph of Section 2.3 to mention the rationale behind switching to L40Z12@600% in the actuation section.

In order to study the electric programming performance, conductive carbon black was incorporated to obtain the electrical conductivity. However, when half percent of lignin was replaced by carbon black, the strain at break of the composite L20C20Z12@0% decreased to around 300%. To avoid the failure of the material, we chose 300% as the mechanical training strain. Larger percent of carbon black would lead to stiffer material and lose the characteristics of artificial muscle material, lower percent of carbon black could not achieve satisfactory conductivity. Thus, L20C20Z12@300% was selected to study the electric-induced actuation performance. Now the reason behind choosing this specific composition of L20C20Z12@300% in the electric programming and actuation section is added in Supplementary Fig. 16 in the

SI document.

11. Page 22 – “The self-strengthening effect and the strain-adaptive stiffening behavior of L20C20Z12@300% were similar as L40Z12@600% and could mimic the mechanical performance of skeletal muscles.”

The authors may have forgotten to add the data supporting this claim.

Response: Thanks. The supporting data was provided in Supplementary Fig. 16 in the SI document.

REVIEWERS' COMMENTS

Reviewer #1 (Remarks to the Author):

The authors have addressed both reviewers' comments in a satisfactory manner.

Reviewer #2 (Remarks to the Author):

My comments have been satisfactorily addressed in great detail by the authors. Thanks.